# Improving the Detection of Epidemic Clones in *Candida parapsilosis* Outbreaks by Combining MALDI-TOF Mass Spectrometry and Deep Learning Approaches

**DOI:** 10.3390/microorganisms11041071

**Published:** 2023-04-20

**Authors:** Noshine Mohammad, Anne-Cécile Normand, Cécile Nabet, Alexandre Godmer, Jean-Yves Brossas, Marion Blaize, Christine Bonnal, Arnaud Fekkar, Sébastien Imbert, Xavier Tannier, Renaud Piarroux

**Affiliations:** 1Groupe Hospitalier Pitié-Salpêtrière, Service de Parasitologie Mycologie, AP-HP, 75013 Paris, France; noshine.mohammad@gmail.com (N.M.); cecile.nabet@aphp.fr (C.N.); jeanyves.brossas@aphp.fr (J.-Y.B.); arnaud.fekkar@aphp.fr (A.F.); renaud.piarroux@aphp.fr (R.P.); 2INSERM, Institut Pierre-Louis d’Épidémiologie et de Santé Publique, Sorbonne Université, 75013 Paris, France; 3CIMI-Paris, Centre d’Immunologie et des Maladies Infectieuses, UMR 1135, Sorbonne Université, 75013 Paris, France; alexandre.godmer@aphp.fr; 4Département de Bactériologie, Hôpital Saint-Antoine, AP-HP, Sorbonne Université, 75012 Paris, France; 5CIMI-Paris, Centre d’Immunologie et des Maladies Infectieuses, CNRS, INSERM, Sorbonne Université, 75013 Paris, France; 6Service de Parasitologie Mycologie, Hôpital Bichat-Claude Bernard, AP-HP, 75018 Paris, France; christine.bonnal@aphp.fr; 7Service de Parasitologie Mycologie, Centre Hospitalier Universitaire de Bordeaux, 33075 Bordeaux, France; sebastien.imbert@chu-bordeaux.fr; 8Sorbonne Université, Inserm, Laboratoire d’Informatique Médicale et d’Ingénierie des Connaissances en e-Santé, LIMICS, 75013 Paris, France; xavier.tannier@sorbonne-universite.fr

**Keywords:** MALDI TOF, epidemiology, *Candida parapsilosis*, neural network, artificial intelligence, outbreak, deep learning

## Abstract

Identifying fungal clones propagated during outbreaks in hospital settings is a problem that increasingly confronts biologists. Current tools based on DNA sequencing or microsatellite analysis require specific manipulations that are difficult to implement in the context of routine diagnosis. Using deep learning to classify the mass spectra obtained during the routine identification of fungi by MALDI-TOF mass spectrometry could be of interest to differentiate isolates belonging to epidemic clones from others. As part of the management of a nosocomial outbreak due to *Candida parapsilosis* in two Parisian hospitals, we studied the impact of the preparation of the spectra on the performance of a deep neural network. Our purpose was to differentiate 39 otherwise fluconazole-resistant isolates belonging to a clonal subset from 56 other isolates, most of which were fluconazole-susceptible, collected during the same period and not belonging to the clonal subset. Our study carried out on spectra obtained on four different machines from isolates cultured for 24 or 48 h on three different culture media showed that each of these parameters had a significant impact on the performance of the classifier. In particular, using different culture times between learning and testing steps could lead to a collapse in the accuracy of the predictions. On the other hand, including spectra obtained after 24 and 48 h of growth during the learning step restored the good results. Finally, we showed that the deleterious effect of the device variability used for learning and testing could be largely improved by including a spectra alignment step during preprocessing before submitting them to the neural network. Taken together, these experiments show the great potential of deep learning models to identify spectra of specific clones, providing that crucial parameters are controlled during both culture and preparation steps before submitting spectra to a classifier.

## 1. Introduction

*Candida parapsilosis* is one of the most common yeasts responsible for human infections. Some studies rank it second just behind *Candida albicans* among the species most frequently responsible for candidemia [1]. Notably, this yeast has been implicated in nosocomial infection epidemics [2], including several outbreaks due to isolates resistant to fluconazole and other azoles, which are the first line of treatment [3,4,5,6]. Furthermore, some of these outbreaks are responsible for high mortality rates in intensive care units, especially if the patients are immunocompromised [7,8]. Recent publications report up to 30% of fluconazole-resistant isolates carrying an A395T mutation (Y132F substitution) in the erg11 gene to explain the observed phenotype. This mutation is likely the main mechanism that confers azole resistance to these isolates. In 2021, our team [9] described an outbreak of *Candida parapsilosis* resistant to fluconazole in the La Pitié Salpêtrière hospital (PSL) in Paris. Two clones infecting mainly ICU patients were identified; one was identified between 2012 and 2017 and the other emerged in 2017 and is unfortunately still active. The worrying spread of these resistant epidemic clones makes it necessary to build appropriate diagnostic tools for detecting clonal resistant isolates among all nonclonal fluconazole-susceptible *C. parapsilosis* identified in the routine flow of our microbiology departments. For now, allocating a given isolate to a clonal set requires the use of molecular methods such as microsatellite typing [10,11] or DNA sequencing. However, these methods are too expensive and time consuming to be implemented as routine activities.

We therefore set out to find a method that would allow clones to be identified directly in the flow of routine analyses without having to implement additional biological assays based on molecular biology. Detecting an epidemiological cluster of drug-resistant microorganisms directly through routine analysis methods would allow microbiologists to alert clinicians, making it possible to rapidly adapt the treatment administered to the patient and thus improve infection management. Currently, matrix-assisted laser desorption/ionization time-of-flight mass spectrometry (MALDI-TOF MS) represents the main routine approach to identify bacteria and yeasts in almost all microbiology laboratories around the world. MALDI-TOF mass spectrometry generates mass spectra corresponding to the main proteins and glycoproteins extracted from microorganisms [12,13,14,15]. The mass spectra can be considered as species-specific fingerprints, allowing accurate identification of purified isolates at the genus and species levels. Recent studies have opened the door for new applications of MALDI-TOF typing approaches with the use of machine learning algorithms. Delavy et al. selected a machine learning model that qualitatively detects fluconazole resistance in the azole-tolerant species *C. albicans* [16]. Most recently, Normand et al. developed a simple deep learning model to identify a clonal population of *Aspergillus flavus* by MALDI-TOF mass spectrometry with a high performance [17]. Unfortunately, unlike the examples cited above, we have quickly discovered that in the case of *Candida parapsilosis*, the protein profiles obtained after MALDI-TOF mass spectrometry were so similar that it was impossible to obtain a good discrimination between the isolates belonging to the resistant clone and others using the model that successively discriminated *Aspergillus flavus* clones.

In this study, we investigated the methods used during the preparation of samples and during the computer analysis of mass spectra to improve the learning phase and, consequently, the discriminatory power of the trained neural network. This study particularly focused on the experimental steps that can influence the performance of epidemic clone identification using deep learning applied to the MALDI-TOF MS spectra. This work constitutes the first effort to analyze the conditions required for the optimal use of MALDI-TOF MS and deep learning in investigating outbreaks in medical mycology. It may be useful for other teams experiencing difficulties in successfully distinguishing between microbial entities with highly similar MALDI-TOF MS patterns.

## 2. Materials and Methods

**Isolates**. Ninety-six isolates that were either susceptible or resistant to fluconazole were selected for this study (Appendix A). Some of the isolates used in the present study have been previously described [9,18]. Among the resistant isolates, 39 belong to the clonal set that recently spread in different ICUs of two hospitals located in Paris (the La Pitié-Salpêtrière hospital (PSL) and Bichat Claude Bernard hospital (BCH)). The remaining isolates were selected from the daily activity of three hospitals (PSL and BCH in Paris and the Pellegrin Hospital in Bordeaux). All *Candida parapsilosis* isolates were cultured in parallel on three types of culture media (Sabouraud Chloramphenicol Gentamycin (SAB-CG; Oxoid, Dardilly, France), Chromagar (CHR; BD, Le Pont de Claix, France) and Blood Agar (BLOOD; BioMérieux, Craponne, France)).

**Genetic diversity**. Microsatellite genotyping was performed as previously described [10]. Briefly, a panel of 6 short tandem repeats was used, resulting in a 12-marker microsatellite profile for each isolate. The resulting microsatellite profiles were then exported and submitted to an unweighted pair group method with arithmetic mean (UPGMA) cluster analysis (Dendro-UPGMA, available at http://genomes.urv.es/UPGMA/, accessed on 12 February 2023) to generate a dendrogram, considering data as categorical values. Isolates with ≥11 identical genotypes by microsatellite typing were grouped and considered to belong to the same clonal set.

**Fluconazole susceptibility**. The minimal inhibitory concentrations of fluconazole were determined by a gradient concentration strip method (Etest; bioMérieux). Isolates were classified as susceptible, intermediate, or resistant according to the EUCAST clinical breakpoints available at http://www.eucast.org/astoffungi/clinicalbreakpointsforantifungals/ (accessed on 12 February 2023).

**MALDI-TOF mass spectra acquisition**. All positive cultures were subjected to the MALDI-TOF MS extraction protocol as previously described by Normand et al. [19] after 24 and/or 48 h of growth. Briefly, *C. parapsilosis* was inactivated in a 70% EtOH solution and proteins were extracted using formic acid and acetonitrile (*v*/*v*). One microliter of protein extract was deposited on one spot of two polished steel targets (two deposits per isolate per culture medium) and covered with one microliter of HCCA matrix. In each experiment, samples from the clone set and from the other category were alternatively deposited to avoid having all clonal spectra in one-half of the target and all nonclonal spectra in the other half. Spectra were acquired using Microflex machines located in four different Parisian laboratories: the mycology department (MYCO-PSL) and the bacteriology department (BACT-PSL) in the Pitié Salpêtrière hospital, the bacteriology department in the Bichat Claude Bernard hospital (BICHAT) and the bacteriology department in Saint-Antoine hospital (SAINT-ANTOINE). For the four mass spectrometers, the default acquisition method (MBT-AutoX) was selected. One default spectrum acquisition parameter from the Flex Control software (version 3.4) was altered as follows: when none of the 800 shots led to a spectrum that met the manufacturer requirements, the sum of the rejected spectra was saved instead of selecting the default option (i.e., do not save), allowing a systematic acquisition of a spectrum, regardless of the wear of the machine. Each deposit identification was checked using the MSI-2 database (https://msi.happy-dev.fr/).


**MALDI-TOF mass spectrometry data analysis.**


**Preprocessing**. MALDI-TOF raw data were preprocessed using python environment 3.8 with smoothing using Fourier transformation, the asymmetric least squares method [20] for the baseline correction and peak picking with the detection of the sign changes in the spectra derivative [21] (Figure 1).

**Alignment**. Alignment of the spectra was performed after the preprocessing step with MSIWarp, a Python package provided with C++ implementation. MSIWarp is a flexible tool compatible with multiple instrument types to perform mass alignment of mass spectrometry imaging spectra [22]. The alignment approach works on TOF data and reduces the mass range shift by applying a recalibration function on mass (*m*/*z*) data and by maximizing a similarity score that considers both the intensity and *m*/*z* position of peaks matched between two spectra. It can be applied using a reference spectrum. Here, the chosen reference spectrum was the spectrum with the highest correlation coefficient with all other spectra.

**Machine learning**. To further differentiate clone and nonclone spectra, a deep learning method involving ANN was implemented with TensorFlow 2.7.0. It was composed of a convolutional part and a fully connected part (architecture typical of a convolutional neural network (CNN)). The classifier (Figure 2) was a very simple CNN [23] model taking a spectrum of 18,000 values as input (this accepts the preprocessed spectra and passes them on to the remaining network). The convolutional block was used to assist in the detection of patterns. It was composed of several layers (3 filters and a kernel size of 6): a convolutional layer to extract the characteristics, a max-pooling layer to reduce and pass on the main information [24] (pool size = 100) and a flatten layer followed by two fully connected layers (512 and 1024 units). A rectified linear unit function (ReLU) [25] was used in the convolutional and fully connected layers as the activation function. Classification was then performed with a normalization layer [26] to improve the class score with a final dense layer of dimension 2, followed by a softmax [27] function to produce the prediction probability over the 2 output classes (clones/others). The learning rate was set by default to 0.001 and the maximum number of epochs was set to 50 with early stopping with patience = 20. We used the Adam optimizer and the categorical cross-entropy loss [28]. The batch size was set to 60.

The preprocessed spectra were fed into a neural network with a convolutional layer, a max-pooling layer and a flattening layer to extract the main features and reduce captured information. The flattened layer was used as a transition to two fully connected layers to optimize classification. Features from the previous layer were then normalized by the normalizing layer followed by the output layer to produce results. The shape of the output layer was added with the batch size, n, set to 1 to simplify the illustration.

**MALDI-TOF mass spectrometry data analysis cross-validation**. For each test, the isolates were divided into five equally sized sets using random selection to preserve the clone/nonclone distribution. We validated our classification system using a nested cross-validation (CV) technique stratified by clone/nonclone classification. Each CV fold was made of a training set composed of the data from 80% of the spectra depending on the criteria tested and a test set comprising the remaining 20% of the spectra. In total, 20 folds were performed with strict separation between the training and the test set, both in terms of isolates, culture media, age of the culture and mass spectrometers. On each fold, the clone/others classification system was trained on the training set and validated on the test set.

**Evaluation metrics**. For each impact assessment, we used the accuracy (percentage of correct identifications), the F1-score, which is a synthesis score used in machine learning, the recall (sensitivity) and the specificity. Confidence intervals at the 95% confidence level were computed using the empirical bootstrap method [29].
$$\mathrm{Accuracy}=\frac{\mathrm{TP}+\mathrm{TN}}{\mathrm{TP}+\mathrm{TN}+\mathrm{FP}+\mathrm{FN}}$$$$\mathrm{Specificity}=\frac{\mathrm{TN}}{\mathrm{TN}+\mathrm{FP}}$$$$\mathrm{Precision}\;\left(\mathrm{PPV}\right)=\frac{\mathrm{TP}}{\mathrm{TP}+\mathrm{FP}}$$$$\mathrm{Recall}\;\left(\mathrm{Sensitivity}\right)=\frac{\mathrm{TP}}{\mathrm{TP}+\mathrm{FN}}$$$$F1\mathrm{score}=2*\frac{\mathrm{Precision}*\mathrm{Recall}}{\mathrm{Precision}+\mathrm{Recall}}$$
where TP are true positives, FP are false positives, TN are true negatives, FN are false-negatives and PPV is the positive predictive value.

**Study design**. The study was designed in four steps (Figure 3). First, using all the spectra that were acquired after 24 h of growth on the three culture media, we compared the machine effect. We used spectra obtained with three of the four machines for the learning phase and we tested the neural network with spectra obtained with the fourth machine. Second, using the same process, we applied the MSIWARP alignment method prior to the learning and testing phases. Third, to test the effect of the culture medium, we detailed the results depending on the culture media used for the growth of the isolates. Finally, using two of the four machines, we acquired the spectra from the 96 isolates at 24 and 48 h of growth to assess the impact of the age of the culture.

**Ethical considerations**. This study was carried out in accordance with the Declaration of Helsinki. The current study was not considered a study involving humans according to French law No. 2012-300, as no clinical or identifying data were used. All the strains were stored anonymously in the Pitié Salpêtrière Hospital Mycology Laboratory.

## 3. Results

### 3.1. Genetic Diversity

Among the 96 selected isolates, 39 were closely related and belonged to our set of clones that we called R2 (Figure 4 and Appendix A). A total of 37 of the 39 isolates corresponded to two widespread clones varying by only seven repeats on one of the alleles (26 isolates with the R2a profile (3A: 28-28; 3B: 49-82; 3C: 48-51; 6A: 8-8; 6B: 7-7; 6C: 7-7) and 11 isolates with the R2b profile (3A: 28-28; 3B: 49-75; 3C: 48-51; 6A: 8-8; 6B: 7-7; 6C: 7-7)). Two other isolates appeared to be close to those two clones and varied in the 3B microsatellite: isolate 307 (3A: 28-28; 3B: 49-76; 3C: 48-51; 6A: 8-8; 6B: 7-7; 6C: 7-7) and isolate 329 (3A: 28-28; 3B: 46-82; 3C: 48-51; 6A: 8-8; 6B: 7-7; 6C: 7-7).

### 3.2. Fluconazole Susceptibility

Among our 96 isolates, 51 showed resistance to fluconazole (minimum inhibitory concentration (MIC) ≥ 4 mg/L) and 45 isolates were susceptible or intermediate to fluconazole (MIC < 4 mg/L) (Appendix A). All of the isolates belonging to the R2 set of clones were resistant to fluconazole, and 38/39 showed resistance ≥ 256 mg/L. Only R2 isolate 329 had an MIC of 16 mg/L s. Most isolates of the R2a profile (25/26) were from the La Pitié-Salpêtrière Hospital between November 2017 and October 2020, while one isolate of this clone was identified at Bichat-Claude Bernard Hospital in June 2021. Isolates of the R2b profile were detected starting in February 2020 in La Pitié-Salpêtrière and May 2021 at Bichat-Claude Bernard. The outbreaks of *C. parapsilosis* from both profiles are still ongoing in the two hospitals.

### 3.3. MALDI-TOF Mass Spectrometry Data Analysis

A total of 2258 spectra were acquired and used for determining the machine, alignment and culture medium impacts, and 768 new spectra were acquired for the assessment of the impact of the age of the culture.

### 3.4. Impact of the Machine and of the Alignment with MSIWarp

The results of the CV realized for each of the four machines are compiled in Table 1. Without alignment, the performance of the CNN model showed results ranging from 68% to 89% for the mean accuracy, depending on the tested machine. This experiment shows that all machines were not equal, even though the acquisition parameters were the same. MSIWarp alignment significantly improved the performance of the CNN model, especially for the two machines that showed lower performances without alignment. Notably, the recall (sensitivity) of the BACT-PSL and BICHAT machines improved from 0.30 to 0.64 and from 0.29 to 0.75, respectively, both without a loss in specificity.

### 3.5. Impact of the Culture Medium per Machine

Keeping the alignment with MSIWarp, we compared the results obtained on the three culture media per machine (Table 2). Depending on the culture medium and the machine tested, the mean accuracy of the CNN model ranged from 0.77% to 0.96%. Except for the MYCO-PSL machine, for which the sensitivity was equivalent in the three culture media, greater performances were obtained on Sabouraud-GC for an equivalent specificity.

### 3.6. Impact of the Age of the Culture on Sabouraud Medium

Keeping the alignment with MSIWarp, we compared the performances obtained after 24 h and 48 h of growth on two machines (MYCO-PSL and SAINT-ANTOINE). Spectra from both machines were pooled and CV was performed only on the age of the culture (Table 3). When considering the same ages of the culture for training and testing, the performances were found to be equal, regardless of the metric taken into account (>90%). When the ages of the culture were crossed, especially when the CNN was trained with spectra from cultures grown for 48 h and tested with spectra from cultures grown for 24 h, the performance was disastrous, with all spectra identified as nonclones.

## 4. Discussion 

Artificial intelligence includes the field of machine learning, which is the development of mathematical algorithms capable of solving problems based on learning from data samples. In this regard, deep learning algorithms (DLs), which use artificial neural networks (ANNs), are a subset of machine learning. In microbiology, particularly in the detection of antimicrobial resistance, these techniques have provided interesting insights [30]. These ANNs are a set of interconnected neurons that are capable of classifying output data from input signals. There are a number of different architectures that can be used, including convolutional neural networks (CNNs), which are known to be very powerful in image recognition [31]. For example, these algorithms have demonstrated their utility in microbiology for automated Gram stain reading [32].

However, in contrast to image recognition, experimental data on MALDI-TOF mass spectra remain scarce. Although MALDI-TOF mass spectrometry has become the main method used for the routine identification of bacteria, yeasts and filamentous fungi, only a few studies have explored the benefit of deep learning algorithms in MALDI-TOF spectra classification. This observation can be applied either in studies designed to distinguish closely related species or to identify a particular characteristic within a microbial species, such as resistance to certain antimicrobial molecules or belonging to an epidemic clone. To date, no study has focused on the preanalytical steps involved in classifying spectra using a neural network. We show here that these steps are important by highlighting the role of the culture media, the growth time, the machine used to acquire the spectra and, finally, the mathematical treatment applied to the spectrum, in particular its alignment with a reference spectrum before its classification by the neural network.

The result can be excellent, mediocre or disastrous depending on whether these parameters are controlled. Thus, a learning process performed on two machines (MYCO-PSL and SAINT ANTOINE) from colonies grown 48 h on Sabouraud-GC agar allowed us to correctly classify 94% of the spectra acquired following the same conditions, while trying to classify spectra acquired after 24 h of growth using the same trained neural network led to disastrous results (all spectra were classified as nonclones). Our results also show that this pitfall could be circumvented by including the two culture times in the learning process, making it possible to obtain a satisfactory classification of the isolates after both 24 and 48 h of culture. The impact of the age of the culture on the shape of the spectrum has already been observed in studies designed to assess the identification performances in medical microbiology, especially for dermatophytes [33,34]. In some cases, this has led to the inclusion of spectra acquired at various ages of the culture in the reference databases to improve the identification performances. In the special case of the search for clones within a yeast species, the degree of precision makes it essential to control this parameter.

Beyond the colony’s time of growth, our study showed the importance of the culture medium on which the colonies are grown in obtaining the most reliable results. This was not a surprise for us, as this parameter has often been pointed out in studies, even though those studies concluded that the impact of such variation on identification reliability was not a hurdle. In the case of the search for clones, the level of precision is such that it would be better to consider this parameter. Our study shows that classifying clones was possible either by extending the learning process to several culture media or by restricting the use of the model to spectra obtained from isolates cultured on the same medium as that used for learning.

The same conclusions can be drawn about the machines used for the learning phase and for the tests. In a previous study on *Aspergillus flavus* clonal detection, we highlighted a machine effect for the learning and testing phases and pointed out difficulties in obtaining satisfactory results with one of the tested machines (BACT-PSL) that was overused [17]. Nevertheless, we show here that by using several machines in the learning phase (leading to an increase in spectra analyzed), it was possible to obtain a satisfactory classification of the spectra for another machine, with 81 to 91% of correctly classified spectra, depending on the machine used to test the model. However, to obtain these results, classification by the neural network should be preceded by an alignment step of the spectra to minimize the variability of the spectra from one machine to the other. Fortunately, such a step can be performed automatically and only takes a fraction of a second for each new spectrum tested on the trained model. Quite unexpectedly, we were able to observe that our neural network could very easily identify the machine on which the spectra had been acquired and the culture medium on which the colony had been grown.

Altogether, these results show that it is possible to use deep neural networks to carry out epidemiological studies at a local level or even on several centers, provided that some parameters are monitored. On the basis of the research carried out in this study, we recommend that any center searching for specific clones in the context of the local spread of an outbreak should perform the learning phase using locally acquired spectra and then test the subsequent model using the same Maldi-ToF mass spectrometer. In addition, the conditions, i.e., culture medium and culture time, under which the colonies were obtained must be identical between the learning and the test phases. In the event that the spectra to be tested are expected to correspond to various acquisition conditions (for example, use of several culture media or several mass spectrometers), we recommend taking into account these conditions in the learning phase. The high impacts of parameters such as culture media or time of growth have also been observed with infrared spectrometry and bacterial typing [35], for which it is recommended to run all samples to be typed in the same experiment. Here, we show that it was possible to obtain satisfactory results when learning and testing were not performed at the same time or on the same machine. This is an interesting finding that needs to be highlighted. The other notable advantage is that a technology commonly used in biomedical laboratories was used as a starting point, which was not the case with the infrared spectrometry study.

However, our study has limitations. First, the number of tested isolates (96, 39 of which corresponded to an outbreak) is low. This certainly restricted the learning abilities of our neural network, as it is well known that the more elements that are included in the learning phase, the better the results are. However, outbreaks occurring in hospital settings usually involve a limited number of cases, especially those involving fungal agents; hence, there is a need to develop approaches suitable for helping with epidemiological investigations as soon as the outbreak is discovered and when the number of cases is still low. Thus, an outbreak involving 39 cases in two different hospitals is already a problem, which is why it is necessary to establish good detection tools.

All isolate identifications in our study were confirmed by MALDI TOF mass spectrometry using both the Bruker database and MSI-2 online, and all obtained *C. parapsilosis* identifications with high scores, confirming the species. Nevertheless, we acknowledge that the MALDI-TOF, even with a high score, may not be enough to ensure the quality of the identification results. Therefore some of the isolates used for this study have been sent to the Belgian collection of microorganisms (IHEM 28980; IHEM 28981; IHEM 28982; IHEM 28983; IHEM 28984; IHEM 28985; IHEM 28986; IHEM 28987; IHEM 28988; IHEM 28989; IHEM 28990; IHEM 28991; IHEM 28992; IHEM 28993). For those 14 isolates now in collection, MALDI-TOF identification was confirmed. We believe that they can be considered as positive controls for our experiment. We did not use an outgroup for our microsatellite experiment nor for our neural network. Indeed, we used very specific microsatellites primers that could not match with any other *Candida spp.*. Hence, the phylogenic tree could not include such outgroups and the very essence of supervised deep learning requires excluding outliers. Including an outgroup in the neural network risks giving uninterpretable results.

In this study, we did not explore the possibilities that artificial intelligence approaches different from deep neural networks could provide (such as support vector machine, PLS discriminant analysis, K nearest neighbors or random forest). We also did not try to develop more sophisticated deep neural networks (recurrent neural network, Siamese neural network, etc.). The focus of this article was rather to explore the different steps preceding the learning phase, as those steps are often overlooked in publications on the matter.

## 5. Conclusions

This study should be considered as a proof of concept aiming to highlight the issues in the use of deep learning methods with MALDI-TOF mass spectrometry for the differentiation of clonal strains from non-clonal strains in an epidemic context. The study focuses on the variations that can lead to misidentification by deep learning in the experimental phase prior to acquisition of the spectra. These are crucial elements to integrate into our knowledge in order to build a neural network model that is robust to these constraints. That being said, nothing prevents microbiologists from using a two-step sequential approach when investigating outbreaks of a certain magnitude. Firstly, the use of CNNs could make it possible to identify the strains potentially related to the epidemic and secondly, confirmation molecular methods could be implemented to confirm a strain belongs to the epidemic clone. In such cases, it is of importance to ensure that the CNN model is sensitive enough for detecting clonal strains.

Overall, the optimization of MALDI-TOF mass spectrum preparation before classification using deep learning techniques is a newly emerging subject, and much remains to be explored on this topic. However, with this study, we demonstrate that such optimization may enhance deep learning results and should eventually allow pushing the limits of MALDI-TOF mass spectrometry. This may open the way to further improvements in the diagnosis of fungal and bacterial outbreaks as a complement to molecular methods.

## Figures and Tables

**Figure 1 microorganisms-11-01071-f001:**
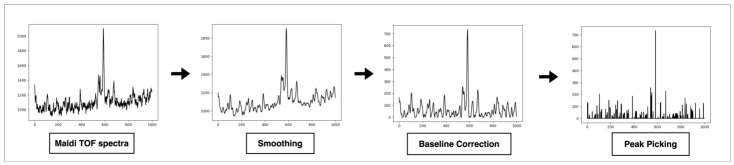
Step-by-step preprocessing of the spectra from the raw spectrum to the processed spectra before use in the machine learning phase.

**Figure 2 microorganisms-11-01071-f002:**
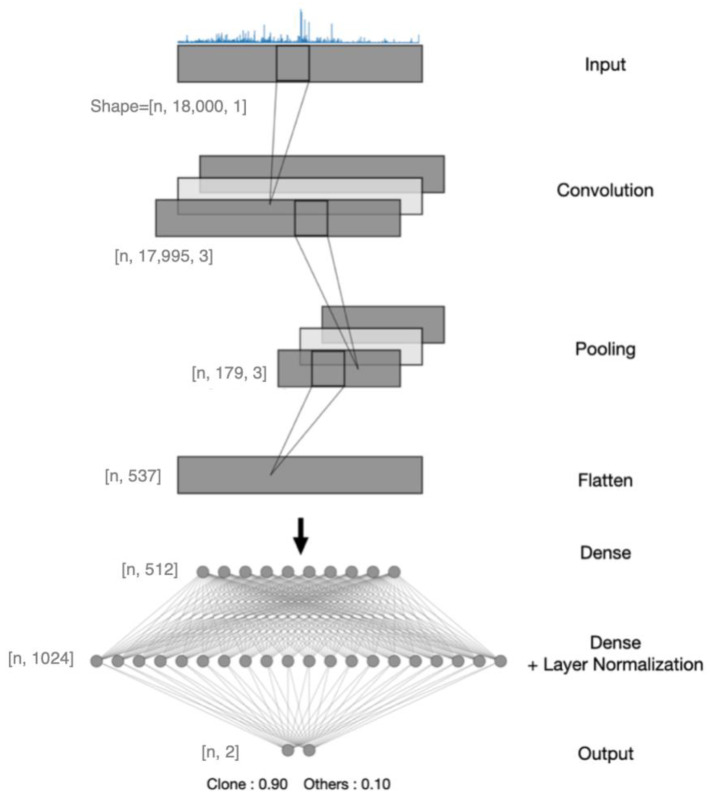
Architecture of the CNN model created and trained with a dataset.

**Figure 3 microorganisms-11-01071-f003:**
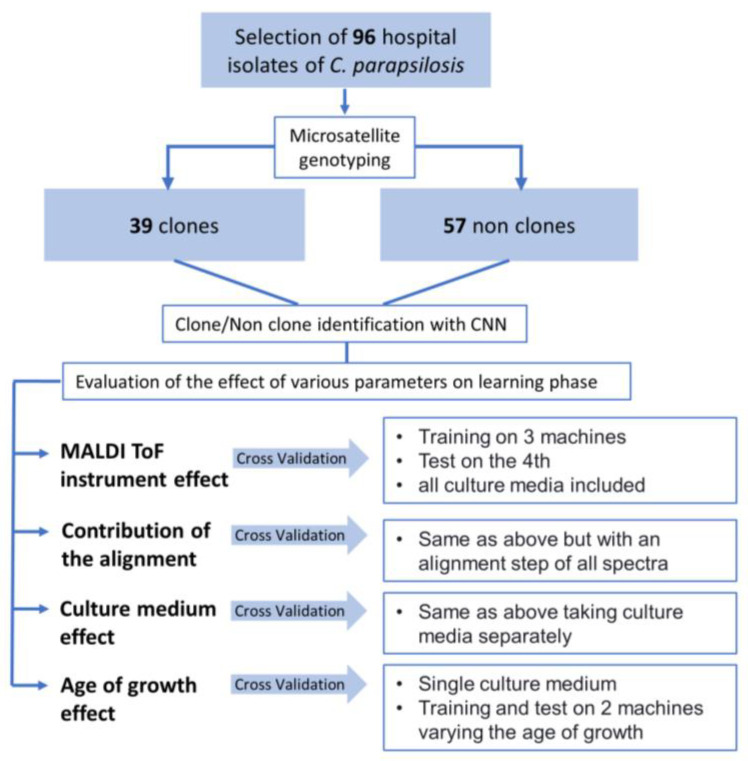
Study design flowchart.

**Figure 4 microorganisms-11-01071-f004:**
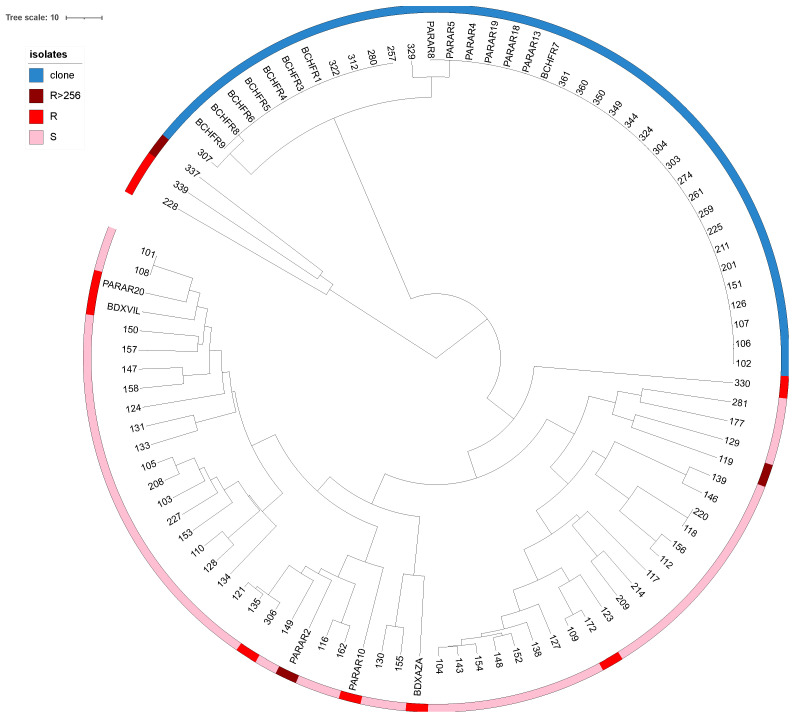
Dendrogram of the 96 *Candida parapsilosis* isolates selected for this study and typed by a microsatellite approach.

**Table 1 microorganisms-11-01071-t001:** Impact of the machine and of the alignment with MSIWarp. Performance of the identification of isolates belonging to the set of clones by the CNN model (cross-validation on five folds). Mean training sets of 1355 spectra obtained on three machines and mean testing set of 113 spectra obtained on the fourth machine.

	Accuracy	F1-Score	Recall (Sensitivity)	Specificity
Machine Tested; Performance Without Alignment				
MYCO-PSL	0.89 [0.87, 0.92]	0.87 [0.84, 0.90]	0.90 [0.86, 0.94]	0.86 [0.83, 0.90]
BACT-PSL	0.70 [0.66, 0.73]	0.44 [0.37, 0.51]	0.30 [0.24, 0.37]	0.95 [0.93, 0.97]
SAINT-ANTOINE	0.83 [0.80, 0.86]	0.74 [0.71, 0.80]	0.65 [0.59, 0.71]	0.90 [0.87, 0.93]
BICHAT	0.68 [0.65, 0.72]	0.42 [0.36, 0.50]	0.29 [0.24, 0.35]	0.88 [0.84, 0.91]
Machine Tested; Performance With Alignment				
MYCO-PSL	0.91 [0.88, 0.93]	0.89 [0.86, 0.92]	0.92 [0.88, 0.95]	0.88 [0.84, 0.91]
BACT-PSL	0.81 [0.78, 0.85]	0.74 [0.68, 0.78]	0.64 [0.57, 0.70]	0.92 [0.89, 0.95]
SAINT-ANTOINE	0.91 [0.89, 0.93]	0.89 [0.86, 0.92]	0.92 [0.88, 0.95]	0.87 [0.83, 0.90]
BICHAT	0.84 [0.81, 0.87]	0.79 [0.75, 0.83]	0.75 [0.68, 0.80]	0.87 [0.84, 0.91]

**Table 2 microorganisms-11-01071-t002:** Impact of the culture medium per machine. Performance of the identification of isolates belonging to the set of clones by the CNN model (cross-validation on five folds). Mean training sets of 452 spectra obtained on three machines and mean testing set of 38 spectra obtained on the fourth machine.

	Accuracy	F1-Score	Recall (Sensitivity)	Specificity
Machine Tested With Alignment; Performance On Chromagar				
MYCO-PSL	0.91 [0.86, 0.94]	0.89 [0.83, 0.94]	0.88 [0.80, 0.95]	0.83 [0.75, 0.90]
BACT-PSL	0.77 [0.71, 0.83]	0.62 [0.55, 0.75]	0.47 [0.37, 0.59]	0.96 [0.92, 0.99]
SAINT-ANTOINE	0.88 [0.83, 0.92]	0.84 [0.78, 0.90]	0.80 [0.71, 0.88]	0.90 [0.84, 0.95]
BICHAT	0.81 [0.75, 0.86]	0.72 [0.64, 0.82]	0.61 [0.50, 0.71]	0.96 [0.91, 0.99]
Machine Tested With Alignment; Performance On Sabouraud-CG				
MYCO-PSL	0.88 [0.84, 0.93]	0.86 [0.80, 0.92]	0.84 [0.75, 0.91]	0.89 [0.83, 0.94]
BACT-PSL	0.93 [0.89, 0.96]	0.92 [0.87, 0.96]	0.95 [0.89, 0.99]	0.88 [0.82, 0.94]
SAINT-ANTOINE	0.89 [0.84, 0.93]	0.89 [0.83, 0.93]	0.95 [0.89, 0.99]	0.84 [0.76, 0.90]
BICHAT	0.89 [0.85, 0.94]	0.88 [0.82, 0.92]	0.89 [0.81, 0.95]	0.92 [0.87, 0.97]
Machine Tested With Alignment; Performance On Blood Agar				
MYCO-PSL	0.92 [0.88, 0.96]	0.89 [0.86, 0.96]	0.87 [0.79, 0.94]	0.95 [0.90, 0.98]
BACT-PSL	0.86 [0.81, 0.90]	0.81 [0.74, 0.88]	0.73 [0.63, 0.83]	0.97 [0.94, 1.00]
SAINT-ANTOINE	0.93 [0.89, 0.96]	0.92 [0.87, 0.96]	0.96 [0.91, 1.00]	0.87 [0.81, 0.93]
BICHAT	0.96 [0.94, 0.99]	0.95 [0.91, 0.99]	0.93 [0.87, 0.99]	0.87 [0.81, 0.93]

**Table 3 microorganisms-11-01071-t003:** Impact of the age of the culture. Performance of the identification of isolates belonging to the set of clones by the CNN model (cross-validation on five folds). L = training set; T = testing set. A total of 307 spectra were used per age of the culture for the training set, while 77 spectra were used per age of the culture for the testing set.

	Accuracy	F1-Score	Recall (Sensitivity)	Specificity
Age of the culture tested with alignment; Performance by ages of the culture				
L24 h/T24 h	0.92 [0.90, 0.95]	0.91 [0.87, 0.94]	0.91 [0.86, 0.94]	0.95 [0.92, 0.98]
L48 h/T48 h	0.94 [0.91, 0.96]	0.92 [0.89, 0.95]	0.93 [0.89, 0.97]	0.95 [0.92, 0.97]
L(24 h + 48 h)/T(24 + 48 h)	0.93 [0.91, 0.94]	0.91 [0.88, 0.93]	0.91 [0.87, 0.94]	0.96 [0.94, 0.97]
Age of the culture tested with alignment; Performance by mixed ages of the culture				
L24 h/T48 h	0.70 [0.65, 0.74]	0.71 [0.65, 0.76]	0.90 [0.85, 0.94]	0.56 [0.50, 0.63]
L48 h/T24 h	0.59 [0.54, 0.64]	0.00 [0.00, 0.00]	0.00 [0.00, 0.00]	1.00 [1.00, 1.00]
L(24 h + 48 h)/T24 h	0.92 [0.90, 0.95]	0.90 [0.86, 0.94]	0.87 [0.82, 0.92]	0.96 [0.93, 0.98]
L(24 h + 48 h)/T48 h	0.91 [0.89, 0.94]	0.89 [0.86, 0.93]	0.89 [0.84, 0.94]	0.93 [0.90, 0.96]

## Data Availability

The data presented in this study are available on request from the corresponding author.

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
