# Peer review of "Improving the Detection of Epidemic Clones in Candida parapsilosis Outbreaks by Combining MALDI-TOF Mass Spectrometry and Deep Learning Approaches"

_microorganisms, 2023, doi:10.3390/microorganisms11041071_

Round 1
Reviewer 1 Report
The paper by Mohammad et al addresses the challenge of identifying fungal clones that spread during hospital outbreaks when current DNA sequencing and microsatellite analysis tools have serious limitations.
The authors propose to use deep learning to analyze mass spectra obtained during routine fungal identification using MALDI-TOF mass spectrometry to differentiate epidemic clones from other isolates. The study focuses on the management of a nosocomial Candida parapsilosis outbreak in two Parisian hospitals and examines the impact of spectra preparation on the performance of a deep neural network to differentiate fluconazole-resistant isolates from a clonal subset of other isolates.
The study reveals the significant impact of culture time, culture media, and machinery on the performance of the classifier and suggests that the inclusion of spectra obtained after 24 and 48 hours of growth during the learning stage can improve the results. The study also demonstrates the potential of including a spectra alignment step during preprocessing to reduce the deleterious effect of machine variability. Finally, the authors highlight the importance of controlling crucial parameters during the cultivation and preparation steps before submitting spectra to the classifier to unlock the potential of deep learning models in identifying clone-specific spectra.
Despite some methodological concerns (pre-analytical factors), the study offers valuable understandings into employing deep learning for tackling the difficulty of identifying fungal clones in hospitals. Given that the domain of enhancing MALDI-TOF mass spectra preparation through deep learning methods is advancing swiftly, the authors' research is likely one of the scarce investigations conducted in this area.
The article is skillfully written, and the methods and materials section, particularly the model section, and the results section are clearly presented.
However, I recommend a few revisions to enhance the manuscript's quality:
1. Consider providing an overall study design flowchart as the text can be confusing, and it is easy to lose track of comparisons. I suggest placing Figure 2 in the supplemental material and adding a new figure in its place to illustrate the study design.
2. In the discussion section, it would be interesting to expand on certain paragraphs:
· Although the authors note that several pre-analytical factors, such as growth time, culture media, and machine, need to be controlled for accurate results, they do not propose a clear strategy for overcoming these limitations.
· The authors propose that satisfactory classification was achieved by increasing the number of spectra and machines used in the learning phase and performing an alignment step of the spectra. It is not clear how many spectra are referenced and whether this should be done with a more homogeneous methodology. This section requires further discussion.
· The authors state that "We also did not try developing more sophisticated deep neural networks." What do they mean by more sophisticated, and how would it make a difference if confounding factors are still present?
Author Response
The different points highlighted by the reviewers are very relevant to better communicate the findings of our study. We have incorporated changes in the manuscript to reflect the suggestions provided by the reviewers. We have highlighted the changes within the manuscript.
The lines indicated in our response to the reviewers correspond to the lines of the revised version (Manuscript-Tracked changes) with the apparent corrections.
Comments from Reviewer 1:
- Consider providing an overall study design flowchart as the text can be confusing, and it is easy to lose track of comparisons. I suggest placing Figure 2 in the supplemental material and adding a new figure in its place to illustrate the study design.
We agree with this comment and included the flowchart in line 226, immediately after the general description of the study and the listing of the different learning processes. We however kept figure 2 in the manuscript as it can be informative for the reader.
- In the discussion section, it would be interesting to expand on certain paragraphs:
- Although the authors note that several pre-analytical factors, such as growth time, culture media, and machine, need to be controlled for accurate results, they do not propose a clear strategy for overcoming these limitations.
Agree. As you have rightly pointed out, we note that several pre-analytical factors, such as growth time, culture media, and machine, need to be controlled for accurate results. We therefore added the following sentences to the discussion: “On the basis of the research carried out in this study, we recommend that any center searching for specific clones in the context of the local spread of an outbreak should perform the learning phase using locally acquired spectra and then test the subsequent model using the same Maldi-ToF mass spectrometer. In addition, the conditions, i.e. culture medium and culture time, under which the colonies were obtained, must be identical between the learning and the test phases. In the event that the spectra to be tested are expected to correspond to various acquisition conditions (for example, use of several culture media or several mass spectrometers), we recommend taking into account these conditions in the learning phase.” This addition can be found on lines 371 to 379 of the revised version of the manuscript.
- The authors propose that satisfactory classification was achieved by increasing the number of spectra and machines used in the learning phase and performing an alignment step of the spectra. It is not clear how many spectra are referenced and whether this should be done with a more homogeneous methodology. This section requires further discussion.
Thank you for pointing this out. We made some changes in order to clarify our point (as stated in the first response, see paragraph above). Note that the number of spectra to be included in the training set in order to obtain an accurate performance for the identification of clones and non-clones depends on the scope of the training. If we limit ourselves to one culture medium, one Maldi ToF mass spectrometer and one culture time, satisfactory results can be achieved even when including a relatively low number of spectra in the learning phase. By contrast, if we expect to test spectra corresponding to diverse acquisition conditions, there is a need for a larger set of spectra for machine learning in order to take all the possible conditions in account. Provided that spectra corresponding to every condition is included in the training set, it is very likely that better identification performances will be obtained by adding more spectra during the learning phase.
Moreover, we decided to amend the sentence in the manuscript in order to better express our meaning. Line 359 now reads: “Nevertheless, we show here that by using several machines in the learning phase (leading to an increase of spectra analyzed), it was possible to obtain a satisfactory classification of the spectra for another machine, with 81 to 91% of correctly classified spectra, depending on the machine used to test the model.”
- The authors state that "We also did not try developing more sophisticated deep neural networks." What do they mean by more sophisticated, and how would it make a difference if confounding factors are still present?
In fact, as you point out, the sentence is not entirely clear. When we used the term "sophisticated", we were referring to the type of neural networks that are more recent than the 1-dimensional convolutional neural network. We meant Siamese neural network type models or even other types of neural networks such as recurrent neural networks. Following your comment, we have taken care to add examples in the sentence in question to facilitate a good understanding of this section. This has led us to re-write “We also did not try developing more sophisticated deep neural networks.” at line 387-388 by “We also did not try developing more sophisticated deep neural networks (Recurrent neural network, Siamese neural network, etc).” Other types of neural networks, such as those mentioned above, would possibly be less sensitive to variations in the experimental parameters, or being more efficient at identifying clones and non-clones. This has to be checked in future experiments.
Reviewer 2 Report
In this paper, the authors evaluate the importance of using mathematical algorithms in the rapid classification of different isolates of Candida parapsilosis starting from protein spectra obtained by MALDI-TOF MS.
The manuscript is well structured and written. However, although the importance of preparing the MALDI-TOF mass spectra has been studied, the results still appear inconclusive, probably because the analyzed isolates are few, as stated in the conclusions. Molecular methods still proved to be "irreplaceable" in the diagnosis of fungal and bacterial epidemics.
Having said that, I believe that this study provides important new experimental data therefore for me the paper can be accepted in its current form.
Author Response
The different points highlighted by the reviewers are very relevant to better communicate the findings of our study. We have incorporated changes in the manuscript to reflect the suggestions provided by the reviewers. We have highlighted the changes within the manuscript.
The lines indicated in our response to the reviewers correspond to the lines of the revised version (Manuscript-Tracked changes) with the apparent corrections.
Comments from Reviewer 2:
The manuscript is well structured and written. However, although the importance of preparing the MALDI-TOF mass spectra has been studied, the results still appear inconclusive, probably because the analyzed isolates are few, as stated in the conclusions. Molecular methods still proved to be "irreplaceable" in the diagnosis of fungal and bacterial epidemics.
You have raised an important point here. As you mentioned, the results still need to be improved. This study should be considered as a proof of concept aiming to highlight the issues in the use of deep learning methods with MALDI-TOF mass spectrometry for the differentiation of clonal strains from non-clonal strains in an epidemic context. The study focuses on the variations that can lead to misidentification by deep learning in the experimental phase prior to acquisition of the spectra. These are crucial elements to integrate into our knowledge in order to build a neural network model that is robust to these constraints. That being said, nothing prevents microbiologists from using a two steps sequential approach when investigating outbreaks of a certain magnitude. Firstly, the use of CNNs could make it possible to identify the strains potentially related to the epidemic, and then secondly, confirmation molecular methods could be implemented to confirm the belonging of a strain to the epidemic clone. In such case, it is of importance to ensure that the CNN model is sensitive enough for detecting clonal strains.
Even though it was not asked by the reviewer, this comment has been added to the conclusion from line 405 to line 416.
Having said that, I believe that this study provides important new experimental data therefore for me the paper can be accepted in its current form.
We thank the reviewer for their comment and for their support regarding the publication of our work.